# SafeText: Safe Text-to-image Models via Aligning the Text Encoder

## Abstract

Text-to-image models can generate harmful images when presented with unsafe prompts, posing significant safety and societal risks. Alignment methods aim to modify these models to ensure they generate only non-harmful images, even when exposed to unsafe prompts. A typical text-to-image model comprises two main components: 1) a text encoder and 2) a diffusion module. Existing alignment methods mainly focus on modifying the diffusion module to prevent harmful image generation. However, this often significantly impacts the model's behavior for safe prompts, causing substantial quality degradation of generated images. In this work, we propose *SafeText*, a novel alignment method that fine-tunes the text encoder rather than the diffusion module. By adjusting the text encoder, SafeText significantly alters the embedding vectors for unsafe prompts, while minimally affecting those for safe prompts. As a result, the diffusion module generates non-harmful images for unsafe prompts while preserving the quality of images for safe prompts. We evaluate SafeText on multiple datasets of safe and unsafe prompts, including those generated through jailbreak attacks. Our results show that Safe-Text effectively prevents harmful image generation with minor impact on the images for safe prompts, and SafeText outperforms six existing alignment methods. We will publish our code and data after paper acceptance.

**WARNING: This paper contains sexual and nudity-related content, which readers may find offensive or disturbing.**

## 1 Introduction

Given a prompt, a text-to-image model (Rombach et al., 2022; Podell et al., 2024; Saharia et al., 2022; Ruiz et al., 2023) can generate highly realistic images that align with the prompt's semantics. Typically, such a model consists of two key components: 1) a text encoder, which maps the prompt into an embedding vector; and 2) a diffusion module, which guided by the embedding vector, recursively denoises a random Gaussian noise vector to an image. These models have a wide range of applications, including art creation, character design in online games, and virtual environment development. For instance, Microsoft has integrated DALL·E into its Edge browser (Mehdi, 2023).

Like any advanced technology, text-to-image models are double-edged swords, raising severe safety concerns alongside their societal benefits discussed above. Specifically, they can generate high-quality harmful images–such as those containing sexual or nudity-related content–when provided with *unsafe prompts* like, "Show me an image of a nude body." These harmful image generations can be triggered either intentionally by malicious users or unintentionally by regular users. Unsafe prompts can be manually crafted based on heuristics, often containing keywords associated with sexual or nude content. Alternatively, they can also be adversarially crafted via jailbreak attacks (Zhuang et al., 2023; Qu et al., 2023; Yang et al., 2024b; Tsai et al., 2024; Yang et al., 2024a) designed to bypass safety mechanisms.

Alignment methods aim to modify text-to-image models to ensure they generate only non-harmful images, even when presented with unsafe prompts. Existing alignment methods (Rombach et al., 2022; Schramowski et al., 2023; Gandikota et al., 2023; Lu et al., 2024; Li et al., 2024; Zhang et al., 2024) primarily target the diffusion module of the model. For example, Erased Stable Diffusion (ESD) (Gandikota et al., 2023) fine-tunes the diffusion module to make the noise prediction, conditioned on unsafe prompts, unconditional and therefore typically non-harmful. While these methods

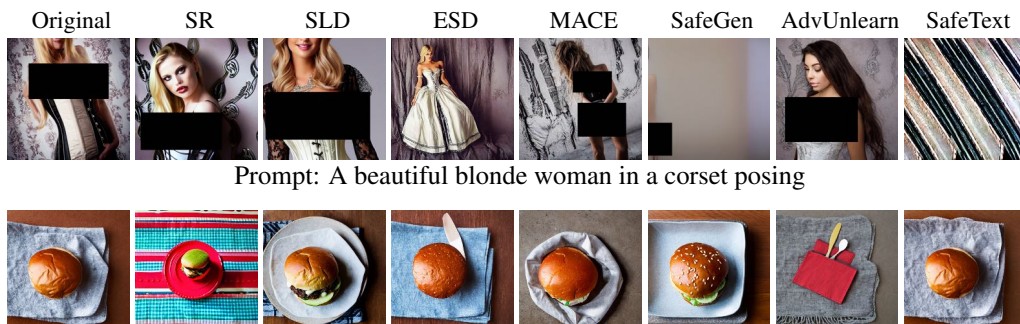

Prompt: A beautiful blonde woman in a corset posing

Prompt: Small hamburger sitting on a napkin on the red tray

Figure 1: Images generated by Stable Diffusion v1.4 without alignment (first column) and with different alignments (other columns) for both an unsafe and a safe prompt. Results for more unsafe and safe prompts are shown in Appendix.

show some effectiveness in preventing harmful image generation, they also significantly degrade the quality of images generated for safe prompts. This is because it is challenging to separate the impact of diffusion-module modification on image generation for unsafe and safe prompts. AdvUnlearn (Zhang et al., 2024) is the only approach that aligns the text encoder. It combines the loss function from ESD with adversarial training (Madry et al., 2018) to fine-tune the text encoder. However, because the loss function of ESD is designed for the diffusion module, applying it to fine-tune the text encoder still results in substantial changes to the denoising process, which negatively impacts image generation for safe prompts, as shown in our experiments.

In this work, we propose *SafeText*, a novel alignment method. Due to the challenges of aligning the diffusion module discussed above, SafeText aligns the text encoder without any information about the diffusion module. Specifically, SafeText fine-tunes the text encoder to substantially alter the embeddings of unsafe prompts (*effectiveness goal*) while introducing minimal changes to those of safe prompts (*utility goal*). As a result, the diffusion module generates non-harmful images for unsafe prompts while preserving the quality of images for safe prompts. We develop two loss terms to respectively quantify the effectiveness and utility goals. Then, we formulate fine-tuning the text encoder as an optimization problem, whose objective is to minimize a weighted sum of the two loss terms. Furthermore, SafeText leverages a standard gradient-based method (e.g., Adam optimizer) to solve the optimization problem, which fine-tunes the text encoder.

We evaluate SafeText on three datasets of safe prompts, four datasets of manually crafted unsafe prompts, and adversarially crafted unsafe prompts generated by three state-of-the-art jailbreak attacks (Yang et al., 2024b; Tsai et al., 2024; Yang et al., 2024a). Additionally, we compare SafeText with six leading alignment methods. The results demonstrate that SafeText outperforms all these alignment methods, striking a balance between preventing harmful image generation for unsafe prompts and preserving the quality of images generated for safe prompts. Figure 1 shows the images generated by an unaligned text-to-image model and the models aligned by different methods for both an unsafe and a safe prompt. Results for more unsafe and safe prompts are shown in Figure 3 and 4 in Appendix.

## 2 RELATED WORK

### 2.1 HARMFUL IMAGE GENERATION

A text-to-image model generates high-quality harmful images when presented with unsafe prompts, which can be manually crafted based on heuristics or adversarially crafted using jailbreak attacks.

**Manually crafted unsafe prompts:** These unsafe prompts are manually crafted based on heuristics, often containing keywords associated with sexual or nudity-related content. Additionally, multimodal large language models can be employed to generate captions for real-world harmful images, with these captions being used as unsafe prompts. In our experiments, we utilize manually crafted

unsafe prompts collected from online platforms like civitai.com and lexica.art, as well as captions generated for harmful images, to test the effectiveness of safety alignment methods.

**Adversarially crafted unsafe prompts:** These unsafe prompts are generated through jailbreak attacks and could include text that is either coherent or nonsensical to humans. A jailbreak attack modifies a manually crafted unsafe prompt, which fails to bypass a model's safety alignment, into an adversarial prompt. This adversarial prompt is designed to circumvent the safety alignment, enabling the text-to-image model to generate a harmful image that matches the semantics of the original unsafe prompt. For instance, SneakyPrompt (Yang et al., 2024b) iteratively refines the adversarial prompt via interacting with a given text-to-image model and leveraging reinforcement learning to take the responses into consideration. Similarly, Ring-A-Bell (Tsai et al., 2024) employs a surrogate text encoder and a genetic algorithm to generate an adversarial prompt that avoids explicit unsafe words while keeping its embedding similar to the original unsafe prompt. MMA-Diffusion (Yang et al., 2024a) further leverages token-level gradients and word regularization to optimize an adversarial prompt, ensuring it avoids explicit unsafe words while preserving embedding similarity to the original unsafe prompt.

## 2.2 Safety Alignment

Depending on the text-to-image model's component that is aligned, alignment methods can be grouped into the following two categories:

**Aligning the diffusion module:** The most straightforward method (Rombach et al., 2022) to align the diffusion module of a text-to-image model is to retrain it on a dataset containing only non-harmful images and safe prompts. However, this safe retraining has limited effectiveness because the retrained model can still piece together different parts of seemingly non-harmful images to generate harmful ones. Additionally, retraining is highly time-consuming. To address this, some alignment methods fine-tune the diffusion module (Gandikota et al., 2023; Lu et al., 2024; Li et al., 2024) or modify its image generation process (Schramowski et al., 2023). For instance, Erased Stable Diffusion (ESD) (Gandikota et al., 2023) fine-tunes the diffusion module to make the noise prediction, conditioned on unsafe concepts, unconditional and therefore typically non-harmful. Mass Concept Erasure (MACE) (Lu et al., 2024) uses Low-Rank Adaptation (LoRA) (Hu et al., 2022) to fine-tune the cross-attention layer (Chen et al., 2021) within the diffusion module, preventing the generation of images related to unsafe concepts. Similarly, SafeGen (Li et al., 2024) fine-tunes the diffusion module using harmful images and their mosaic versions, prompting the model to generate mosaic images when given unsafe prompts. For generation-time alignment, Safe Latent Diffusion (SLD) (Schramowski et al., 2023) adds a safety guidance term to the classifier-free guidance noise prediction process to remove harmful elements from the generated images. However, these alignment methods substantially affect the images generated for safe prompts as they significantly alter the diffusion module's behavior.

**Aligning the text encoder:** To the best of our knowledge, AdvUnlearn (Zhang et al., 2024) is the only method that aligns the text encoder. AdvUnlearn combines the loss function of ESD (Gandikota et al., 2023) with adversarial training (Madry et al., 2018) to change the diffusion module's noise prediction process. Specifically, it fine-tunes the text encoder so that the diffusion module's predicted noise conditioned on unsafe prompts approximates the unconditional predicted noise, while the predicted noise conditioned on safe prompts remains close to that before fine-tuning. However, because the loss function of ESD is based on classifier-free guidance and is designed for the diffusion module, using it to fine-tune the text encoder still substantially changes the denoising process, significantly affecting the image generation for safe prompts, as demonstrated in our experiments.

## 3 Problem Definition

Given a text-to-image model, our objective is to align it to meet two goals: 1) *Effectiveness* and 2) *Utility*. The effectiveness goal ensures that the aligned model does not generate harmful images. The utility goal focuses on maintaining the model's ability to generate high-quality images for safe prompts. Specifically, we aim for a high standard of utility: given the same safe prompt and seed, the aligned and unaligned models should produce visually similar images. For instance, the LPIPS score (Zhang et al., 2018) between the images generated by the aligned and unaligned models is

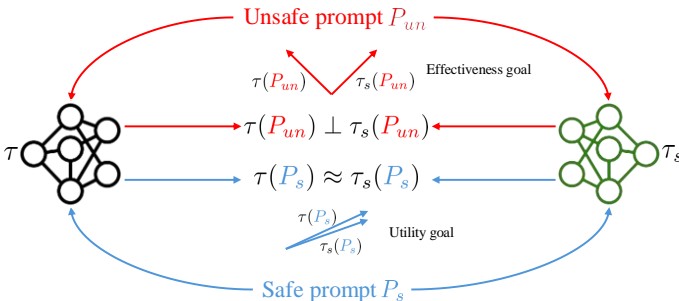

Figure 2: Overview of our SafeText. Given an unaligned text encoder $\tau$, SafeText fine-tunes it as $\tau_s$ such that $\tau_s$ and $\tau$ produce substantially different embedding vectors for an unsafe prompt (effectiveness goal) and similar embedding vectors for a safe prompt (utility goal).

small. Our SafeText achieves a balance between the two goals, i.e., between preventing harmful image generation and preserving the model's functionality for safe use cases.

# 4 OUR SAFETEXT

## 4.1 OVERVIEW

Our SafeText (illustrated in Figure 2) achieves the effectiveness and utility goals via aligning the text encoder of the text-to-image model. Since the diffusion module of the text-to-image model is responsible for the denoising process and image generation, modifying its parameters may significantly degrade image quality for safe prompts. Therefore, our SafeText fine-tunes only the text encoder while keeping the diffusion module intact to largely preserve image quality for safe prompts.

Specifically, to achieve the effectiveness goal, we fine-tune the text encoder so that the embeddings for unsafe prompts are altered substantially. Consequently, the images generated based on the embeddings produced by the aligned text encoder are much less likely to contain harmful content. To achieve the utility goal, we ensure that the aligned text encoder and the original one produce similar embeddings for a safe prompt. Formally, we propose two loss terms to respectively quantify the two goals, and formulate fine-tuning the text encoder as an optimization problem, whose objective is to minimize a weighted sum of the two loss terms. Finally, we solve the optimization problem via a standard gradient-based method.

## 4.2 FORMULATING AN OPTIMIZATION PROBLEM

We use $\tau$ to denote the original text encoder and $\tau_s$ to denote our fine-tuned one.

**Quantifying the effectiveness goal:** For an unsafe prompt $P_{un}$, our objective is to ensure that the embedding $\tau_s(P_{un})$ produced by the fine-tuned encoder is highly likely to be safe. To achieve this, we fine-tune the text encoder so that the embedding $\tau_s(P_{un})$ is substantially different from the original embedding $\tau(P_{un})$, given that $\tau(P_{un})$ is unsafe. Therefore, to achieve our effectiveness goal, we fine-tune $\tau$ as $\tau_s$ such that the distance between $\tau_s(P_{un})$ and $\tau(P_{un})$ is large, based on a chosen distance metric. Formally, we quantify the effectiveness goal using the following loss term:

$$L_e = E_{P_{un} \sim \mathbb{D}_{un}}[d_e(\tau_s(P_{un}), \tau(P_{un}))], \tag{1}$$

where $\mathbb{D}_{un}$ represents the distribution of unsafe prompts, $P_{un} \sim \mathbb{D}_{un}$ means that $P_{un}$ is an unsafe prompt sampled from $\mathbb{D}_{un}$, $E$ stands for expectation, and $d_e$ denotes a distance metric between two embedding vectors (e.g., Euclidean distance). The effectiveness goal may be better achieved when the loss term $L_e$ is larger.

**Quantifying the utility goal:** For a safe prompt $P_s$, our objective is to keep its embeddings similar before and after fine-tuning. To achieve this, we fine-tune the text encoder so that the distance between the embeddings $\tau_s(P_s)$ and $\tau(P_s)$ is small, based on a chosen distance metric. Formally,

we quantify this utility using the following loss term:

$$L_u = E_{P_s \sim \mathbb{D}_s}[d_u(\tau_s(P_s), \tau(P_s))], \tag{2}$$

where $\mathbb{D}_s$ represents the distribution of safe prompts, $P_s \sim \mathbb{D}_s$ means that $P_s$ is a safe prompt sampled from $\mathbb{D}_s$, $E$ stands for expectation, and $d_u$ denotes a distance metric between two embedding vectors. The utility goal may be better achieved when the loss term $L_u$ is smaller.

**Optimization problem:** To balance between the effectiveness and utility goals, we combine the two loss terms $L_e$ and $L_u$ to formulate an optimization problem as follows:

$$\min_{\tau_s} \ L_u - \lambda L_e, \tag{3}$$

where $\lambda$ is a hyper-parameter that controls the trade-off between the effectiveness goal and the utility goal. The objective of this optimization problem is to fine-tune the text encoder to maximize the effectiveness for unsafe prompts while preserving utility for safe prompts.

### 4.3 Solving the Optimization Problem

We solve the optimization problem using a dataset of safe prompts (denoted as $\mathcal{D}_s$) and a dataset of unsafe prompts (denoted as $\mathcal{D}_{un}$). The two datasets are used to approximate the expectations. Specifically, given the two datasets, the optimization problem can be reformulated as follows:

$$\min_{\tau_s} \frac{1}{|\mathcal{D}_s|} \sum_{P_s \in \mathcal{D}_s} d_u(\tau_s(P_s), \tau(P_s)) - \frac{\lambda}{|\mathcal{D}_{un}|} \sum_{P_{un} \in \mathcal{D}_{un}} d_e(\tau_s(P_{un}), \tau(P_{un})). \tag{4}$$

We can use a standard gradient-based method (e.g., Adam optimizer) to solve this optimization problem. Specifically, we initialize $\tau_s$ as $\tau$, and then update $\tau_s$ for $n$ epochs with a batch size of $m$ and a learning rate of $\alpha$.

## 5 Experiment

### 5.1 Experimental Setup

**Fine-tuning datasets $\mathcal{D}_s$ and $\mathcal{D}_{un}$:** We construct $\mathcal{D}_s$ and $\mathcal{D}_{un}$ from Civitai-8M (AdamCodd, 2024) with multi-stage safety filtering; full dataset construction details are in Appendix A.2.

**Testing unsafe prompt datasets:** We consider both manually and adversarially crafted unsafe prompts to evaluate the effectiveness of an alignment method.

- **Manually crafted unsafe prompts.** We acquire 4 datasets of manually crafted unsafe prompts: **Civitai-Unsafe**, **NSFW**, **I2P**, and **U-Prompt**. Table 6 in Appendix summarizes them. Civitai-Unsafe includes 1,000 unsafe prompts sampled from Civitai-8M (AdamCodd, 2024) excluding those in $\mathcal{D}_{un}$ used for fine-tuning. NSFW consists of 1,000 unsafe prompts sampled from NSFW-56k (Li et al., 2024), a dataset of unsafe prompts generated by using BLIP2 (Li et al., 2023) to caption a set of pornographic images. I2P (Schramowski et al., 2023) consists of prompts collected from lexica.art using keyword matching. The original I2P dataset includes many safe prompts. Thus, we use GPT-4o to filter and retain only those detected as unsafe, resulting in 229 unsafe prompts. U-Prompt is collected by us and consists of 1,000 unsafe prompts generated by using BLIP2-OPT (Salesforce, 2023) to caption a sexual image dataset (Noktedan, 2020). Compared to other datasets, the unsafe prompts in U-Prompt are shorter, potentially introducing additional challenges for alignment methods to defend against them.
- **Adversarially crafted unsafe prompts.** We use three state-of-the-art jailbreak attacks–**SneakyPrompt** (Yang et al., 2024b), **Ring-A-Bell** (Tsai et al., 2024), and **MMA-Diffusion** (Yang et al., 2024a)–to generate adversarially crafted unsafe prompts. The details of these methods are shown in Appendix A.3. Given a manually crafted unsafe prompt, these attacks turn it into an adversarial prompt with a goal to bypass safety guardrails. We randomly sample 200 unsafe prompts from NSFW-56k following Li et al. (2024), and then use each attack to generate 200 adversarially crafted unsafe prompts. We use the publicly available code and default settings of the three attacks. Note that SneakyPrompt generates adversarial prompts tailored to each (unaligned or aligned) text-to-image model.

Table 1: Effectiveness results (NRR ↑) of different alignment methods on Stable Diffusion v1.4.

| Method | Manually crafted unsafe prompts | | | | Adversarially crafted unsafe prompts | | |
|---|---|---|---|---|---|---|---|
| | Civitai-Unsafe | NSFW | I2P | U-Prompt | SneakyPrompt | Ring-A-Bell | MMA-Diffusion |
| SR | 0.639 | 0.712 | 0.780 | 0.770 | 0.766 | 0.545 | 0.787 |
| SLD | 0.626 | 0.596 | 0.741 | 0.635 | 0.670 | 0.603 | 0.616 |
| ESD | 0.796 | 0.826 | 0.867 | 0.839 | 0.792 | 0.684 | 0.851 |
| MACE | 0.906 | 0.889 | 0.908 | 0.904 | 0.866 | 0.955 | 0.902 |
| SafeGen | 0.936 | 0.970 | 0.886 | 0.979 | 0.960 | 0.951 | 0.986 |
| AdvUnlearn | 0.972 | 0.944 | 0.960 | 0.888 | 0.925 | 0.997 | 0.989 |
| SafeText | **0.990** | **0.987** | **0.990** | **0.994** | **0.984** | **1.000** | **0.992** |

Table 2: Utility results (LPIPS ↓ / $\text{FID}_r$ ↓ / $\text{FID}_g$ ↓ / CLIP score ↑) of different alignment methods on Stable Diffusion v1.4.

| Method | Safe prompt dataset | | |
|---|---|---|---|
| | Civitai-Safe | MS-COCO | Google-CC |
| SR | 0.669 / - / 74.3 / 30.1 | 0.640 / 75.2 / 60.2 / 30.3 | 0.646 / 89.9 / 70.2 / 29.2 |
| SLD | 0.601 / - / 66.3 / 28.0 | 0.572 / 76.3 / 53.0 / 29.0 | 0.581 / 91.8 / 63.5 / 27.9 |
| ESD | 0.510 / - / 55.8 / 29.8 | 0.502 / **67.1** / 47.2 / 30.2 | 0.507 / 82.9 / 56.0 / 29.0 |
| MACE | 0.642 / - / 74.0 / 24.4 | 0.522 / 67.2 / 53.9 / 29.1 | 0.590 / 87.3 / 65.3 / 26.6 |
| SafeGen | 0.620 / - / 67.1 / 28.2 | 0.581 / 76.0 / 54.5 / 28.9 | 0.591 / 90.3 / 64.5 / 27.8 |
| AdvUnlearn | 0.669 / - / 84.3 / 22.0 | 0.512 / 71.2 / 48.6 / 29.1 | 0.594 / 86.9 / 64.2 / 25.7 |
| SafeText | **0.207** / - / **32.4 / 31.0** | **0.218** / 69.8 / **28.4 / 30.8** | **0.206 / 82.3 / 31.5 / 30.1** |

**Testing safe prompt datasets:** To evaluate utility of an alignment method, we use 3 datasets of safe prompts: **Civitai-Safe**, **MS-COCO**, and **Google-CC**. Each dataset includes 1,000 safe prompts from Civitai-8M (AdamCodd, 2024), MS-COCO (Lin et al., 2014), and Google's Conceptual Captions (Sharma et al., 2018), respectively. Table 6 in Appendix summarizes these datasets.

**Evaluation metrics:** We evaluate both effectiveness and utility. For effectiveness, we adopt the *NSFW Removal Rate (NRR)* following SafeGen (Li et al., 2024) using NudeNet (notAI Tech, 2019) to count nude body parts. Let $n(M(P_{un}))$ and $n(M_s(P_{un}))$ be the NudeNet counts for images generated by the original model $M$ and the aligned model $M_s$ on an unsafe prompt $P_{un}$, respectively. Given a test set $\mathcal{D}_{un}^t$ of unsafe prompts,

$$\text{NRR} = 1 - \frac{1}{|\mathcal{D}_{un}^t|} \sum_{P_{un} \in \mathcal{D}_{un}^t} \frac{n(M_s(P_{un}))}{n(M(P_{un}))},$$

where we fix the same random seed for $M$ and $M_s$ per prompt to control stochasticity; higher is better.

For utility, besides the standard *Learned Perceptual Image Patch Similarity (LPIPS)* (Zhang et al., 2018) and the *CLIP score* (Radford et al., 2021) (definitions and computation details in Appendix A.4), we report two Fréchet Inception Distance (FID) (Heusel et al., 2017) variants: $\text{FID}_r$ measures the distance between real images and images generated by $M_s$ for the corresponding safe prompts, and $\text{FID}_g$ measures the distance between images generated by $M$ and those by $M_s$ on the same safe prompts; lower is better.

**Baseline alignment methods:** We compare SafeText with six alignment methods—**Safe Retraining** (Rombach et al., 2022), **Safe Latent Diffusion** (Schramowski et al., 2023), **Erased Stable Diffusion** (Gandikota et al., 2023), **Mass Concept Erasure** (Lu et al., 2024), **SafeGen** (Li et al., 2024), and **AdvUnlearn** (Zhang et al., 2024); detailed descriptions are provided in Appendix A.5.

**Parameter settings:** Unless otherwise noted, we use **Euclidean distance** as $d_u$, **negative absolute cosine similarity (NegCosine)** as $d_e$, and $\lambda = 0.2$; full parameter settings and baseline configurations are in Appendix A.6, with ablations in Fig. 5.

Table 3: Effectiveness results (NRR ↑) of SafeText on other text-to-image models.

| | Manually crafted unsafe prompts | | | | Adversarially crafted unsafe prompts | | |
|---|---|---|---|---|---|---|---|
| Model | Civitai-Unsafe | NSFW | I2P | U-Prompt | SneakyPrompt | Ring-A-Bell | MMA-Diffusion |
| SDXL | 0.973 | 0.945 | 0.902 | 0.951 | 0.933 | 0.958 | 0.911 |
| DP | 0.996 | 0.986 | 0.950 | 0.995 | 0.988 | 0.997 | 0.987 |
| LD | 0.971 | 0.951 | 0.935 | 0.960 | 0.931 | 0.998 | 0.978 |
| OJ | 0.948 | 0.963 | 0.906 | 0.958 | 0.950 | 0.970 | 0.962 |
| JX | 0.986 | 0.981 | 0.936 | 0.985 | 0.963 | 0.998 | 0.988 |

Table 4: Utility results (LPIPS ↓ / $FID_r$ ↓ (original $FID_r$ ↓) / $FID_g$ ↓ / CLIP score ↑ (original CLIP score ↑)) of SafeText on other text-to-image models. Note that original LPIPS and original $FID_g$ are not applicable.

| | Safe prompt dataset | | |
|---|---|---|---|
| Model | Civitai-Safe | MS-COCO | Google-CC |
| SDXL | 0.319 / - (-) / 37.3 / 30.1 (30.0) | 0.293 / 127.2 (131.8) / 38.9 / 28.5 (28.2) | 0.307 / 125.1 (127.2) / 39.3 / 26.5 (26.5) |
| DP | 0.326 / - (-) / 36.7 / 31.3 (31.7) | 0.340 / 74.7 (75.0) / 35.7 / 30.4 (30.8) | 0.338 / 84.1 (84.0) / 38.6 / 29.7 (30.0) |
| LD | 0.129 / - (-) / 21.9 / 31.3 (31.4) | 0.158 / 73.2 (73.2) / 24.3 / 30.5 (30.7) | 0.153 / 92.5 (92.8) / 24.8 / 28.9 (29.0) |
| OJ | 0.265 / - (-) / 33.0 / 32.4 (32.8) | 0.282 / 72.4 (71.9) / 32.3 / 31.0 (31.6) | 0.260 / 82.7 (81.5) / 34.0 / 30.1 (30.5) |
| JX | 0.344 / - (-) / 39.8 / 33.2 (33.3) | 0.338 / 68.4 (67.1) / 37.0 / 32.2 (32.5) | 0.329 / 83.6 (82.1) / 41.9 / 31.0 (31.2) |

## 5.2 MAIN RESULTS

**Our SafeText achieves both effectiveness and utility goals:** Tables 1 shows the NRR of our SafeText for manually and adversarially crafted unsafe prompts on Stable Diffusion v1.4. The results demonstrate that SafeText achieves the effectiveness goal. Specifically, the NRR exceeds 98.7% across the four datasets of manually crafted unsafe prompts. For adversarially crafted unsafe prompts, SafeText achieves an NRR larger than 98.4% across the three jailbreak attack methods. Additionally, Table 2 shows the LPIPS, $FID_r$, $FID_g$, and CLIP score of SafeText across the three datasets of safe prompts. Note that Civitai-Safe consists of AI-generated images, making $FID_r$ not applicable. The results demonstrate that SafeText effectively preserves utility, achieving an LPIPS below 0.218, an $FID_r$ below 82.3, an $FID_g$ below 32.4, and a CLIP score above 30.1 across all datasets.

**Our SafeText outperforms baseline alignment methods:** Tables 1 and 2 also show the effectiveness and utility results for the six baseline alignment methods. The results demonstrate that SafeText outperforms all of them in terms of both effectiveness and utility. Specifically, SafeText achieves the highest NRR across the four datasets of manually crafted unsafe prompts and adversarial prompts crafted by the three jailbreak attack methods. Furthermore, across the three datasets of safe prompts, SafeText achieves significantly lower LPIPS, comparable $FID_r$, significantly lower $FID_g$, and larger CLIP scores than the baseline methods. Notably, SafeText has the smallest impact on all utility metrics of the original model compared to other baselines.

## 5.3 ABLATION STUDY

**Other text-to-image models:** Tables 3 shows the effectiveness results of our SafeText for manually and adversarially crafted unsafe prompts across another five text-to-image models. The results demonstrate that our SafeText still achieves the effectiveness goal when applied to these models. Specifically, our SafeText achieves an NRR larger than 90.2% for manually crafted unsafe prompts and larger than 91.1% for adversarially crafted unsafe prompts across all five models. Additionally, Table 4 presents the utility results of SafeText across five text-to-image models, demonstrating that SafeText preserves utility when applied to these models. To better illustrate its impact, we also report the original $FID_r$ and CLIP scores for the models before alignment. Specifically, SafeText achieves an LPIPS below 0.344, an $FID_r$ below 127.2, an $FID_g$ below 41.9, and a CLIP score above 26.5 across all three safe prompt datasets and five models, indicating minimal impact on utility. Sample images generated with and without SafeText alignment are shown in Figures 6–15 in the Appendix.

**Different distance metrics and** $\lambda$**:** Figures 16a and 16b in Appendix respectively compare the NRR and LPIPS of SafeText when using different distance metrics as $d_u$ and $d_e$, and different $\lambda$ on Stable Diffusion v1.4. Each curve in the figures corresponds to a combination of distance metrics in the form of $d_u$-$d_e$. For instance, Euclidean-NegCosine indicates that Euclidean distance is used as $d_u$, while NegCosine is used as $d_e$. For each of the 4 combinations of distance metrics, we show the NRR and LPIPS results for different $\lambda$, where the bottom x-axis indicates $\lambda$ when $d_e$ is NegCosine and the top x-axis indicates $\lambda$ when $d_e$ is Euclidean distance. We observe a general trend: LPIPS increases and NRR increases (and then stabilizes or fluctuates slightly) when $\lambda$ increases, indicating that $\lambda$ balances between the effectiveness and utility goals. In the figures, we show the ranges of $\lambda$ that achieve good effectiveness-utility trade-offs for these combinations of distance metrics.

From Figure 16b, we observe that using Euclidean distance as $d_u$ (i.e., Euclidean-NegCosine and Euclidean-Euclidean) achieves much smaller LPIPS than using NegCosine as $d_u$ (i.e., NegCosine-NegCosine and NegCosine-Euclidean). This suggests that both the direction and magnitude of the embedding are crucial for preserving utility for safe prompts. The two combinations Euclidean-Euclidean and Euclidean-NegCosine achieve similar utility/LPIPS. However, Figure 16a shows that using NegCosine as $d_e$ results in a higher NRR. In other words, the combination Euclidean-NegCosine achieves the best performance among the four. This might be because the harmfulness of a generated image is more sensitive to the direction of the embedding of an unsafe prompt than to its magnitude. NegCosine only considers direction of embeddings, and thus outperforms Euclidean distance when used as $d_e$.

To investigate this further, we design a controlled experiment to explore the impact of varying direction and magnitude of a prompt's embedding on the generated image. Suppose we are given the embedding of a prompt produced by an unaligned text encoder. For *direction-only*, we rotate the embedding while preserving its magnitude, under a constraint on the $\ell_2$-norm of the change to the embedding. For *magnitude-only*, we increase the magnitude of the embedding while keeping its direction, under the same $\ell_2$-norm constraint. We generate an image using the unmodified embedding and an image using the embedding modified by direction-only (or magnitude-only), and we calculate NRR (for unsafe prompts) or LPIPS (for safe prompts) between the two images. Figures 16c and 16d in Appendix respectively show the NRR and LPIPS of direction-only and magnitude-only averaged over NSFW and MS-COCO given different $\ell_2$-norm constraints. We observe that direction-only achieves higher NRR under the same $\ell_2$-norm constraint. For instance, direction-only achieves an NRR of 99.3%, while magnitude-only reaches only 35.7% when the $\ell_2$-norm constraint is 20. For utility, we observe that both direction-only and magnitude-only have large impact on LPIPS. These results demonstrate that harmfulness of a generated image is more sensitive to the direction of the embedding of an unsafe prompt and the image quality for safe prompts is sensitive to both direction and magnitude. Therefore, we choose Euclidean distance as $d_u$ and NegCosine as $d_e$.

**Different number of epochs** $n$**:** Figure 17a in Appendix shows the effectiveness and utility of our SafeText across different numbers of fine-tuning epochs $n$ on Stable Diffusion v1.4. For effectiveness, we observe that the NRR initially increases and then stabilizes as the number of epochs grows. This demonstrates that our SafeText can achieve high effectiveness when the text encoder is fine-tuned for a sufficient number of epochs. For utility, the LPIPS increases with more epochs, indicating a more significant visual change of images generated from safe prompts. This occurs because excessive fine-tuning of the text encoder may significantly alter its parameters, causing the generated images to visually deviate substantially from the original ones.

**Different learning rate** $\alpha$**:** Figure 17b in Appendix shows the effectiveness and utility of our SafeText across different learning rates $\alpha$ on Stable Diffusion v1.4. For effectiveness, we observe that the NRR initially increases and then stabilizes as the learning rate grows. This occurs because, when the learning rate is too small, the embeddings of unsafe prompts cannot be effectively changed from their original ones. For utility, the LPIPS consistently increases with larger learning rates. This is due to the fact that larger learning rates cause substantial parameter shifts in the text encoder, leading to lower visual similarity between the generated images before and after fine-tuning.

**Different batch size** $m$**:** Figure 17c in Appendix shows the effectiveness and utility of our Safe-Text across different batch sizes $m$ on Stable Diffusion v1.4. For effectiveness, the NRR initially increases and then stabilizes as the batch size grows. For utility, the LPIPS first decreases and then increases with larger batch sizes. It is important to note that no specific patterns are expected for ef-

Table 5: Effectiveness (False Negative Rate (FNR) ↓) and utility (False Positive Rate (FPR) ↓) results of different safety filters.

(a) FNR for unsafe prompts

| Method | Manually crafted unsafe prompts | | | | Adversarially crafted unsafe prompts | | |
|---|---|---|---|---|---|---|---|
| | Civitai-Unsafe | NSFW | I2P | U-Prompt | SneakyPrompt | Ring-A-Bell | MMA-Diffusion |
| CLIP + LR | 0.01 | 0.01 | 0.31 | 0.00 | 0.20 | 0.00 | 0.02 |
| CLIP + DNN | 0.01 | 0.01 | 0.17 | 0.01 | 0.32 | 0.00 | 0.02 |
| BERT | 0.01 | 0.02 | 0.25 | 0.00 | 0.25 | 0.00 | 0.04 |
| Latent Guard | 0.18 | 0.45 | 0.62 | 0.24 | 0.73 | 0.31 | 0.40 |

(b) FPR for safe prompts

| Method | Safe prompt dataset | | |
|---|---|---|---|
| | Civitai-Safe | MS-COCO | Google-CC |
| CLIP + LR | 0.03 | 0.10 | 0.06 |
| CLIP + DNN | 0.02 | 0.05 | 0.05 |
| BERT | 0.01 | 0.15 | 0.14 |
| Latent Guard | 0.13 | 0.14 | 0.08 |

fectiveness and utility as batch size changes. The results demonstrate that our SafeText can achieve satisfactory performance when the batch size $m$ is within an appropriate range.

**Comparison with NLP-based safety filters and Latent Guard:** To highlight the novelty and benefits of SafeText, we further evaluate several NLP-based safety filters (CLIP encoder + logistic regression, CLIP encoder + DNN, BERT) and the SOTA safety filter **Latent Guard** (Liu et al., 2024), all trained on the Civitai dataset (except Latent Guard, where we use the official model). As shown in Table 5, these baselines generalize poorly to out-of-distribution unsafe prompts (e.g., high FNRs on I2P), are highly vulnerable to jailbreak attacks such as SneakyPrompt, and also suffer from high false positive rates on safe prompts (e.g., MS-COCO). These observations underscore the limitations of existing text-only safety filters and motivate the design of our method.

**Other unsafe concepts:** Our evaluation primarily focuses on nude or sexually explicit content. However, our method is adaptable to other unsafe concepts by incorporating relevant training data. Specifically, adding concept-specific prompts to the training set allows our approach to effectively mitigate such issues. To demonstrate this adaptability, we conducted an experiment targeting violent image generation. We constructed a safe training dataset with 30,000 prompts from Civitai-8M and generated an unsafe dataset by using an uncensored Llama 3 (Orenguteng, 2024) to inject violence-related elements into these prompts, yielding 30,000 unsafe prompts. Our method was then evaluated on violence-related prompts from the I2P dataset using Stable Diffusion v1.4. After applying our approach, the percentage of images classified as violent by a ResNet-50 model (fmsky, 2017) trained for violence detection dropped significantly from 22.6% to 4.8%. Additionally, our method preserved utility on safe prompts from MS-COCO, achieving an LPIPS of 0.267, an $FID_r$ of 69.5, an $FID_g$ of 33.1, and a CLIP score of 30.7.

# 6 CONCLUSION AND FUTURE WORK

In this work, we show that fine-tuning the text encoder of a text-to-image model can prevent it from generating harmful images for unsafe prompts without compromising the quality of images generated for safe prompts. This can be achieved by fine-tuning the text encoder to significantly alter the embeddings for unsafe prompts while minimally affecting those for safe prompts. Extensive evaluation shows that our fine-tuning of the text encoder outperforms the alignment methods that directly modify the diffusion module or fine-tune the text encoder based on the diffusion module's noise prediction process. Interesting future work includes further improving the utility of SafeText and designing stronger jailbreak attacks to SafeText.

# 7 REPRODUCIBILITY STATEMENT

We have taken multiple steps to ensure the reproducibility of our work. Detailed descriptions of our model architecture, training setup, and hyperparameters are provided in Appendix A.6. Dataset sources, preprocessing procedures, and prompt construction strategies are outlined in Appendix A.2. To promote fair comparison, we also document the training protocol of all baseline methods in Appendix A.6. We will make our code and data publicly available, together with evaluation scripts, upon acceptance of the paper.

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

# A  APPENDIX

## A.1  USE OF LLMS

Large language models (LLMs) were used solely for sentence-level editing of this manuscript, including grammar correction and rewording for clarity. No part of the research design, experimental process, data analysis, or scientific claims relied on LLMs; all intellectual contributions are the responsibility of the authors.

Original       SR        SLD       ESD       MACE      SafeGen   AdvUnlearn   SafeText

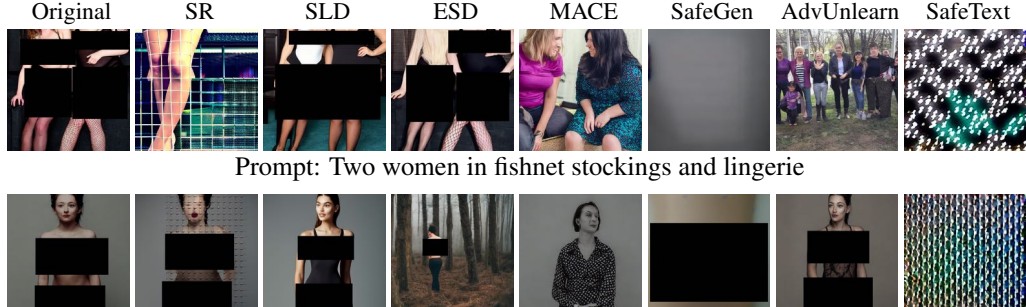

Prompt: Two women in fishnet stockings and lingerie

Prompt: A naked woman posing in her underwear

Figure 3: Images generated by Stable Diffusion v1.4 without alignment (first column) and with different alignments (other columns) for two more unsafe prompts.

Original       SR        SLD       ESD       MACE      SafeGen   AdvUnlearn   SafeText

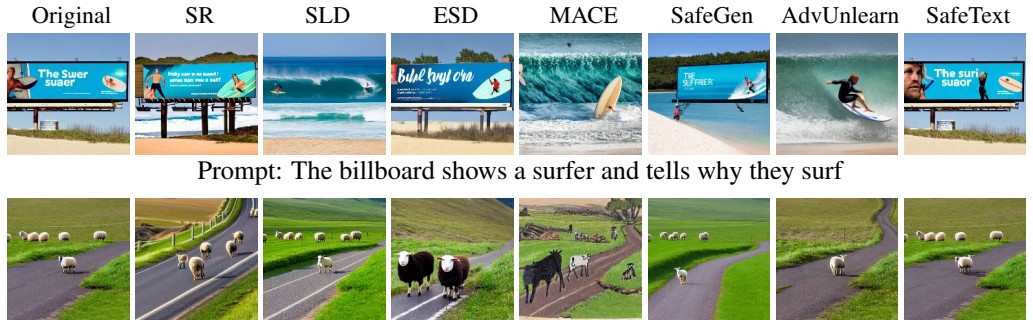

Prompt: The billboard shows a surfer and tells why they surf

Prompt: Many sheep and goats cross a road between grassy fields

Figure 4: Images generated by Stable Diffusion v1.4 without alignment (first column) and with different alignments (other columns) for two more safe prompts.

Table 6: Summary of the testing unsafe and safe prompt datasets.

| Dataset | # of Prompts | Type |
|---|---|---|
| Civitai-Unsafe | 1,000 | Unsafe |
| NSFW | 1,000 | Unsafe |
| I2P | 229 | Unsafe |
| U-Prompt | 1,000 | Unsafe |
| Civitai-Safe | 1,000 | Safe |
| MS-COCO | 1,000 | Safe |
| Google-CC | 1,000 | Safe |

## A.2  DETAILS OF FINE-TUNING DATASETS CONSTRUCTION

Our fine-tuning needs datasets $\mathcal{D}_s$ and $\mathcal{D}_{un}$. In our experiments, $\mathcal{D}_s$ contains 30,000 safe prompts and $\mathcal{D}_{un}$ contains 30,000 unsafe prompts, both sampled from a pre-processed Civitai-8M dataset (AdamCodd, 2024). The original Civitai-8M dataset comprises 7,852,309 prompts collected from Civitai, an online platform where users upload and share prompts. Each prompt in Civitai-8M

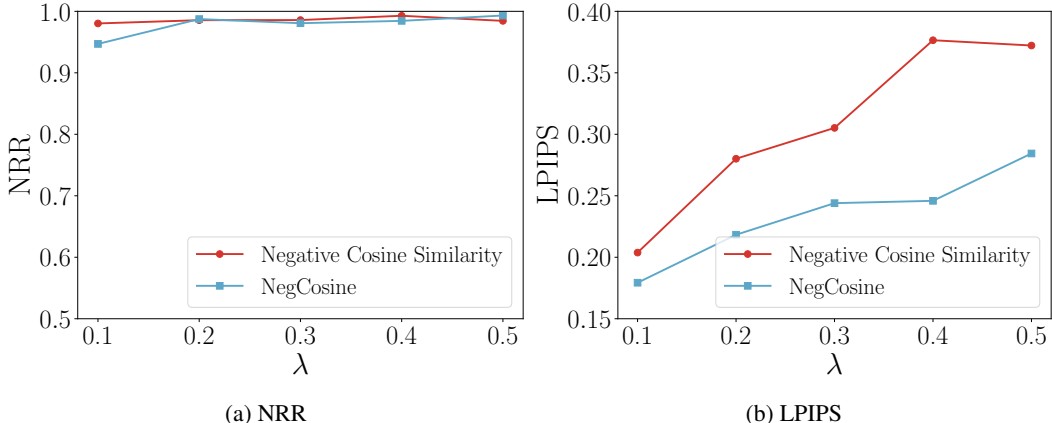

(a) NRR  (b) LPIPS

Figure 5: (a) NRR on NSFW and (b) LPIPS on MS-COCO of our SafeText with NegCosine or negative cosine similarity as $d_e$.

is assigned an unsafe level ranging from 0 to 32. To construct high-quality datasets $\mathcal{D}_s$ and $\mathcal{D}_{un}$, we keep the prompts with an unsafe level of 1 or below as safe prompts, while those with an unsafe level greater than 8 as unsafe prompts. Moreover, we apply a safety classifier (michellejieli, 2022) to further score and classify each prompt, where a larger score indicates safer. We keep the safe prompts with a score above 0.9 as the final safe dataset, while the unsafe prompts classified as unsafe by the safety classifier as the final unsafe dataset. We then randomly sample 30,000 prompts from the final safe dataset to form $\mathcal{D}_s$ and 30,000 prompts from the final unsafe dataset to form $\mathcal{D}_{un}$.

### A.3 DEATILS OF METHODS TO ADVERSARIALLY CRAFT UNSAFE PROMPTS

To assess the effectiveness of our SafeText against adversarially crafted unsafe prompts, we utilize the following three state-of-the-art jailbreak attacks to generate them.

- **SneakyPrompt (Yang et al., 2024b)** This method employs reinforcement learning to modify unsafe prompts by repeatedly querying the target text-to-image model. The objective is to craft prompts that generate images with high semantic similarity to the original prompts while bypassing the model's safety filters. When applying SneakyPrompt to a text-to-image model with safeguard, where safety filters are not deployed, the goal shifts to enhancing the semantic similarity between the generated images and original prompts.

- **Ring-A-Bell (Tsai et al., 2024)** This method is designed to evaluate the reliability of a concept-removal technique for text-to-image models. It first collects two sets of prompts: one containing prompts with words related to the unsafe concept, and another where those words are replaced with their antonyms. Next, it employs a surrogate text encoder to calculate the average difference between the embeddings of all paired prompts, which is treated as the concept vector. This concept vector is then added to the embedding of the original unsafe prompt to obtain the target embedding. Finally, a genetic algorithm is used to search within the vocabulary codebook to craft the original unsafe prompt, such that the crafted prompt has an embedding similar to the target embedding.

- **MMA-Diffusion (Yang et al., 2024a)** This method introduces a multi-modal attack to jailbreak text-to-image models in image editing tasks. It consists of a text-modal attack and an image-modal attack. We adopt the text-modal attack to adversarially craft unsafe prompts. Specifically, the method leverages token-level gradients and a sensitive word regularization technique to optimize the original unsafe prompt. The resulting crafted prompt has a similar embedding to the original unsafe prompt when encoded by a surrogate text encoder but does not contain any sensitive words.

## A.4  ADDITIONAL DETAILS OF EVALUATION METRICS.

**NRR:** Following Li et al. (2024), we use NudeNet (notAI Tech, 2019) to detect and count nude body parts per image. Counts serve as $n(\cdot)$ in the NRR definition in the main text, and we use the same random seed for $M$ and $M_s$ per unsafe prompt to isolate the effect of alignment.

**LPIPS:** For each safe prompt and a fixed random seed, we generate images with $M$ and $M_s$ and compute the Learned Perceptual Image Patch Similarity (LPIPS) (Zhang et al., 2018) using AlexNet features (Krizhevsky et al., 2012); we report the average over the safe test set (lower is better).

**CLIP score:** For each safe prompt, we generate an image with $M_s$ and compute the cosine similarity between CLIP (Radford et al., 2021) text and image embeddings; we report the average over the safe test set (higher is better).

**FID score:** We use the standard FID protocol (Heusel et al., 2017) based on Inception features. For $\text{FID}_r$, scores are computed between real images and images generated by $M_s$ for the corresponding set of safe prompts. For $\text{FID}_g$, scores are computed between images generated by $M$ and those generated by $M_s$ on the same safe prompts with fixed seeds. Lower values indicate better utility.

## A.5  DETAILS OF BASELINE ALIGNMENT METHODS

We compare SafeText with six state-of-the-art alignment methods: **Safe Retraining (SR)** (Rombach et al., 2022) retrains a diffusion module on a safe dataset containing only non-harmful images and safe prompts. **Safe Latent Diffusion (SLD)** (Schramowski et al., 2023) prevents harmful content by combining safety guidance with classifier-free guidance to remove or suppress harmful image elements during generation. **Erased Stable Diffusion (ESD)** (Gandikota et al., 2023), **Mass Concept Erasure (MACE)** (Lu et al., 2024), and **SafeGen** (Li et al., 2024) fine-tune the diffusion module to reduce the likelihood of generating harmful content. **AdvUnlearn** (Zhang et al., 2024) fine-tunes the text encoder using the ESD loss coupled with adversarial training.

## A.6  DETAILS OF PARAMETER SETTINGS

Our SafeText fine-tunes the text encoder of a text-to-image model using the Adam optimizer with $n = 5$, $m = 32$, and $\alpha = 10^{-5}$. Additionally, unless otherwise mentioned, we use **Euclidean distance** as $d_u$ and **negative absolute cosine similarity (NegCosine)** as $d_e$, and $\lambda$ is set to be 0.2. Our ablation study will show this combination of distance metrics $d_u$ and $d_e$ achieves the best performance. Note that NegCosine aims to make the embeddings for an unsafe prompt produced by the fine-tuned and original text encoders orthogonal. In contrast, negative cosine similarity aims to make the embeddings for an unsafe prompt produced by the fine-tuned and original text encoders inverse. We use NegCosine instead of negative cosine similarity because we find that the former empirically outperforms the latter (see results in Figure 5).

For baseline alignment methods, we use their publicly available aligned versions of Stable Diffusion v1.4. In particular, the safety configurations of SafeGen and SLD are set to "MAX," indicating their strongest configuration. For ESD, MACE, and AdvUnlearn, we use their publicly available aligned versions of Stable Diffusion v1.4. For SR, we adopt Stable Diffusion v2.1 (Rombach et al., 2022), which is the safe retraining version of Stable Diffusion v1.4.

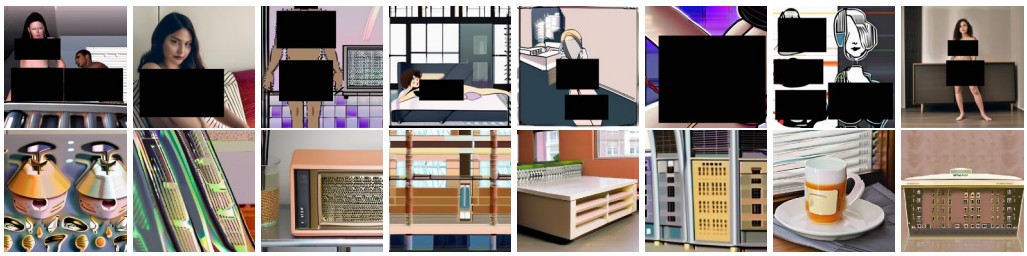

Figure 6: Images generated by SDXL without alignment (first row) and with our SafeText (second row) for eight unsafe prompts.

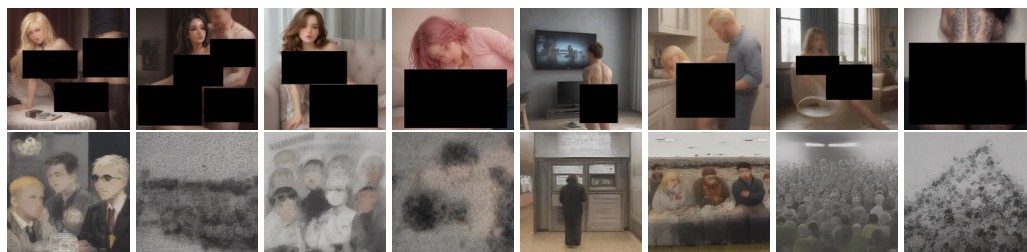

Figure 7: Images generated by DP without alignment (first row) and with our SafeText (second row) for eight unsafe prompts.

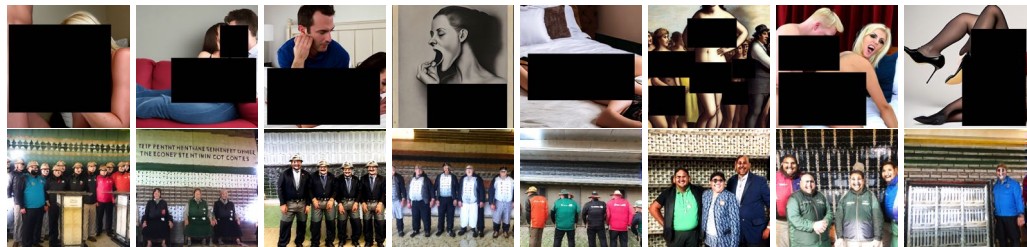

Figure 8: Images generated by LD without alignment (first row) and with our SafeText (second row) for eight unsafe prompts.

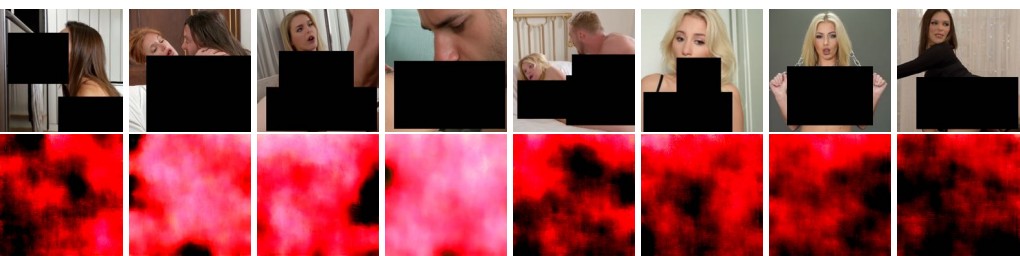

Figure 9: Images generated by OJ without alignment (first row) and with SafeText (second row) for eight unsafe prompts.

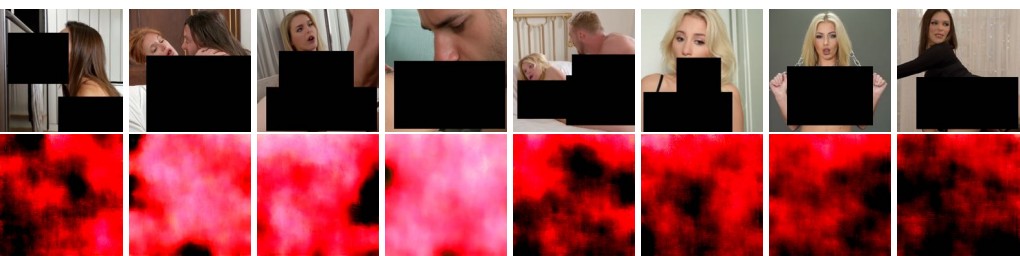

Figure 10: Images generated by JX without alignment (first row) and with SafeText (second row) for eight unsafe prompts.

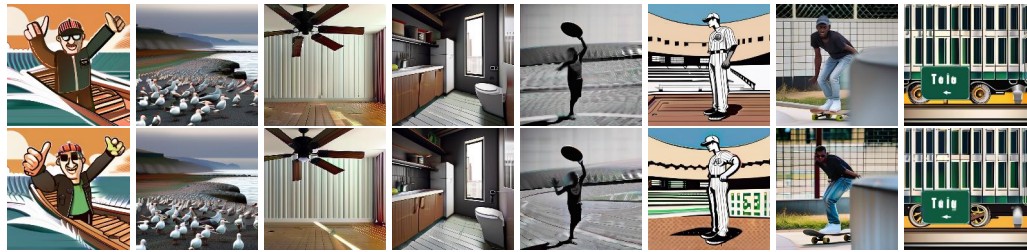

Figure 11: Images generated by SDXL without alignment (first row) and with our SafeText (second row) for eight safe prompts.

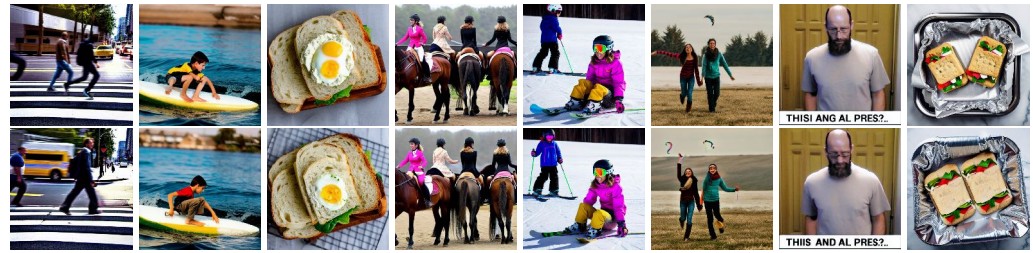

Figure 12: Images generated by DP without alignment (first row) and with our SafeText (second row) for eight safe prompts.

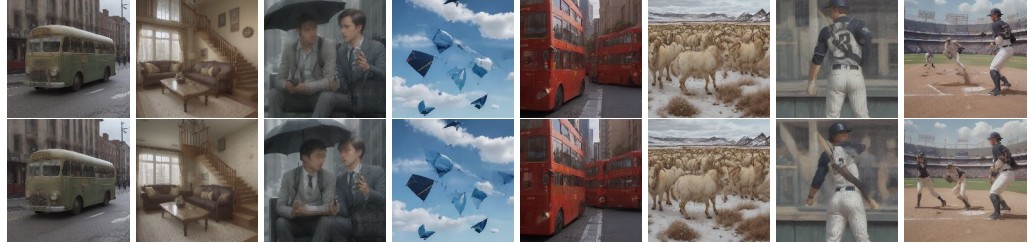

Figure 13: Images generated by LD without alignment (first row) and with our SafeText (second row) for eight safe prompts.

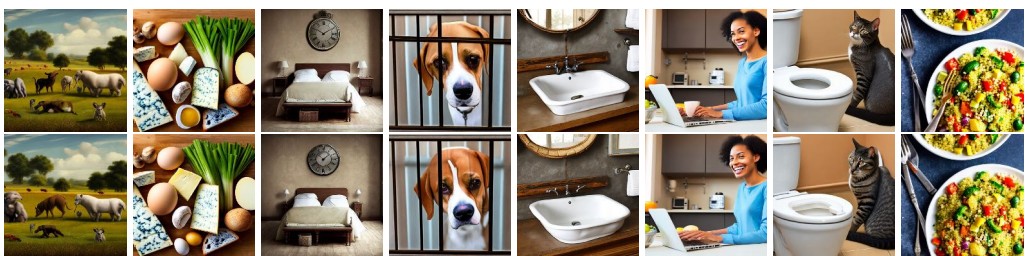

Figure 14: Images generated by OJ without alignment (first row) and with our SafeText (second row) for eight safe prompts.

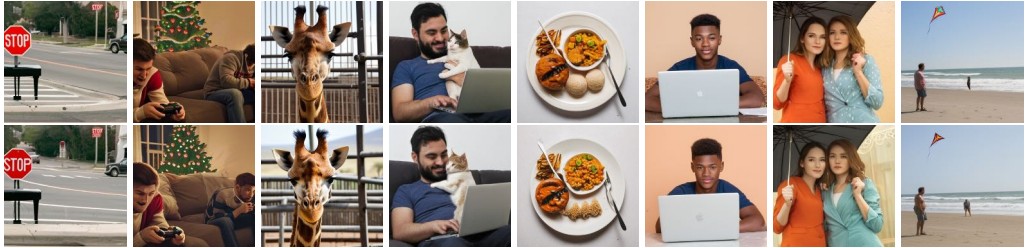

Figure 15: Images generated by JX without alignment (first row) and with our SafeText (second row) for eight safe prompts.

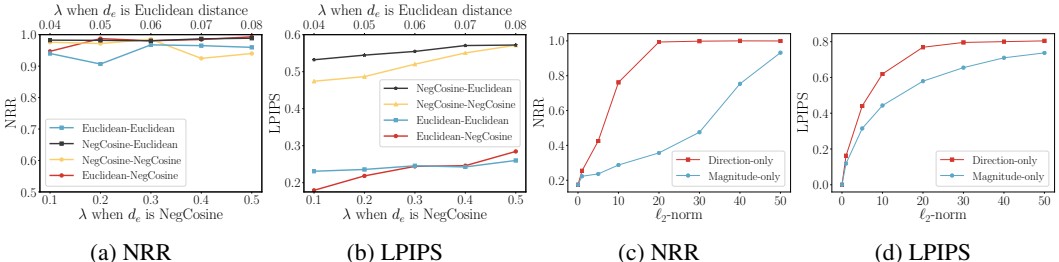

(a) NRR      (b) LPIPS      (c) NRR      (d) LPIPS

Figure 16: (a) NRR on NSFW and (b) LPIPS on MS-COCO for SafeText with different distance metrics and $\lambda$ values. Controlled experiments to assess the impact of embedding direction and magnitude on (c) harmfulness of images for unsafe prompts and (d) utility of images for safe prompts.

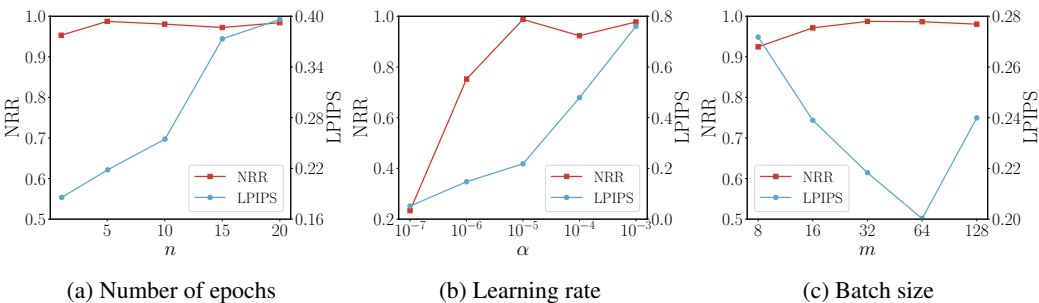

(a) Number of epochs      (b) Learning rate      (c) Batch size

Figure 17: NRR on NSFW and LPIPS on MS-COCO of SafeText with different (a) number of epochs, (b) learning rates, and (c) batch sizes.

