# OpenReview forum: "SafeText: Safe Text-to-image Models via Aligning the Text Encoder"
_ICLR.cc/2026/Conference — ICLR 2026 Conference Withdrawn Submission_

### Official Review · Reviewer_1wBr · 2025-10-26

**Soundness:** 3
**Presentation:** 3
**Contribution:** 2
**Rating:** 2
**Confidence:** 3

**Summary:**

This paper proposes an alignment method to mitigate the generation of harmful images. Specifically, the proposed method modifies the text encoder to adjust the embedding vectors for unsafe prompts while keeping similar embeddings for safe prompts. The evaluation is conducted on both safe and unsafe prompts, where the unsafe ones include manually crafted and adversarially generated prompts. The authors also compare their method with several state-of-the-art baselines, showing competitive performance.

**Strengths:**

- Important topic

- Well written and well structured

- Strong performance

**Weaknesses:**

- Unclear text encoder

- Potential vulnerability to more advanced attacks

- Trade-off between NRR and LPIPS

- Lack of efficiency comparison

**Questions:**

This paper introduces SafeText, a novel method for aligning text-to-image models by fine-tuning the text encoder. The approach is effective at preventing harmful image generation while preserving image quality for safe prompts. Overall, I enjoyed reading this paper. It is very well written, and the theoretical sections are clearly explained.

My main concerns are in the following.

1. Unclear text encoder

While the authors highlight optimizing the text encoder as the main distinction from prior work and a key contribution, the optimization formulation itself seems rather standard. What I find particularly confusing is that the paper never specifies which text encoders are used in the experiments. For instance, SDXL employs two distinct text encoders, and it remains unclear whether the proposed method works equally well for both, or if there are differences in their behavior and effectiveness.

2. Potential vulnerability to more advanced attacks

The optimization framework assumes that unsafe and safe prompts are clearly distinguishable. This allows the optimization to increase the distance between unsafe prompts and their embeddings, thereby reducing their utility, while maintaining the utility for safe prompts. However, as jailbreak techniques evolve, adversarial prompts may become more sophisticated and semantically ambiguous. This could cause the optimization process to inadvertently penalize safe prompts, leading to degraded performance on benign inputs. I wonder how the authors think about this issue. At what point does the proposed method fail when the distinction between unsafe and safe prompts becomes blurred?

3. Choice of hyperparameters

The ablation study reveals a critical trade-off between NRR and LPIPS depending on the number of training epochs. While more epochs improve safety, they degrade image quality. This raises concerns about the practicality of the proposed method, as finding the optimal balance may require careful and potentially expensive model-specific hyperparameter tuning.

4. Cost analysis

The paper provides a comprehensive comparison in terms of safety and utility but omits an analysis of computational efficiency. A discussion of training time, parameter overhead, and inference costs compared to the baselines would be beneficial for assessing the practicality of SafeText for real-world deployment.

Minor: In Table 3, several model abbreviations are used directly, but it is unclear what specific models they refer to. In addition, it is not specified which text encoders are used for these different models, which makes it difficult to interpret the performance differences across them.

Overall, I believe it does not yet meet the standard for acceptance in its current form.

---

### Official Review · Reviewer_NrmM · 2025-10-28

**Soundness:** 3
**Presentation:** 2
**Contribution:** 3
**Rating:** 4
**Confidence:** 3

**Summary:**

This paper proposes SafeText to address the issue where text-to-image models generate harmful images when presented with unsafe prompts. SafeText is a novel alignment technique that fine-tunes the text encoder instead of modifying the diffusion module, as traditional methods do. Its core lies in optimizing two loss terms: the fine-tuned text encoder significantly alters the embedding vectors of unsafe prompts (ensuring the diffusion module generates non-harmful images to achieve the effectiveness goal) while minimizing impacts on the embedding vectors of safe prompts (preserving the quality of generated images to achieve the utility goal). Experiments on multiple datasets demonstrate its effectiveness and adaptability.

**Strengths:**

1. SafeText only aligns the text encoder, resulting in minimal changes to embeddings of safe prompts while substantially altering those of unsafe ones.
2. SafeText achieves high effectiveness while maintaining strong utility, outperforming six baseline methods on Stable Diffusion v1.4.
3. The method is effective against both manually crafted unsafe prompts and adversarially crafted ones generated by state-of-the-art jailbreak techniques.
4. The authors commit to releasing code and data upon paper acceptance, facilitating reproducibility of the research.

**Weaknesses:**

1. The main evaluation focuses on nude/sexually explicit content, with only preliminary verification on violent content. Other unsafe concepts (e.g., hate speech-related images, depictions of dangerous behaviors) lack systematic testing.
2. The method is only tested on Stable Diffusion v1.4; results may not generalize to newer or larger-scale models with distinct text encoders or diffusion module architectures.
3. There is no discussion of computational costs or fine-tuning efficiency in comparison to baseline methods.
4. The paper adopts Euclidean distance for utility loss and negative cosine similarity for effectiveness loss, but lacks in-depth analysis of their unique suitability. It only compares four metric combinations without exploring other potential metrics (e.g., direct cosine similarity) or explaining how metric selection impacts performance across different prompt types. This leaves uncertainty about whether the chosen metrics are universally optimal or merely effective for the tested datasets.

**Questions:**

1. Evaluation on other T2I models.
2. Dicussion of the loss function selection.

---

### Official Review · Reviewer_YfUh · 2025-10-31

**Soundness:** 2
**Presentation:** 2
**Contribution:** 2
**Rating:** 2
**Confidence:** 4

**Summary:**

This paper proposes SafeText, a method that fine-tunes the text encoder of a text-to-image model to prevent the generation of harmful images. The strength of the work lies in its extensive empirical demonstration that this approach achieves a superior trade-off between safety and image quality compared to existing methods. However, from the training objective, changing an unsafe prompt's embedding does not ensure the resulting image is safe, only that it is different from the original  embeddings. Also, the safety evaluation is narrow, relying primarily on a single tool (NudeNet) for a limited class of harm (nudity and violence as an extended scenario).

**Strengths:**

1.  The paper demonstrates a practical advantage of aligning the text encoder over the diffusion module. The extensive experimental results consistently show that the proposed method achieves a superior trade-off, effectively suppressing harmful content while better preserving image quality for safe prompts compared to existing baselines.

2.  The work includes a well-designed ablation study. The analysis of different distance metrics and the controlled experiment investigating the impact of embedding direction versus magnitude provide useful insights.

**Weaknesses:**

1.  From the designed optimization objective function, this work implies that altering the text embedding of an unsafe prompt will lead to a safe image. However, this only ensures
the image generated by altered unsafe-prompt embedding, is different from the original, unsafe one, does not necessarily guarantee that the altered embedding will map to a safe concept.

2.  The evaluation of safety is reliant on NudeNet for detecting sexually explicit content. This is a narrow definition of "harmful" content. While the paper states that other harmful category can be included following the same approach, however, the scalability of this approach is questionable.

3.  While the paper tests against several existing jailbreak attacks, a more robust safety method should be evaluated against an adaptive attacks specifically designed to target SafeText. An adversary aware of SafeText's mechanism could develop attacks that are optimized to find adversarial prompts whose embeddings remain close to the unsafe region even after the encoder's transformation, or that exploit the specific loss function.

4.  It is not clear, how will SafeText compare with a simple, yet well-tuned safety filter that operates on the embeddings of the original, unaltered text encoder. This work would be more compelling if it demonstrated that SafeText provides a clear advantage over simply classifying and blocking prompts based on their embeddings from the original encoder, especially in terms of robustness to jailbreaks and preservation of utility.

**Questions:**

My questions have been stated in the detailed comments.

---

### Official Review · Reviewer_vrhV · 2025-11-01

**Soundness:** 3
**Presentation:** 3
**Contribution:** 3
**Rating:** 4
**Confidence:** 3

**Summary:**

This paper proposes SafeText, a novel method for aligning text-to-image models to prevent the generation of harmful content, specifically focusing on nudity and sexual imagery. The core idea is to fine-tune only the text encoder component of the model, leaving the diffusion module untouched. The method is framed as an optimization problem with two objectives: 1) an "effectiveness" goal to significantly alter the text embeddings of unsafe prompts, and 2) a "utility" goal to minimally change the embeddings of safe prompts. The author, leveraging the concept of contrastive learning, designed a simple loss function to push unsafe embeddings away from their original representations while keeping safe embeddings close. Through extensive experiments on Stable Diffusion and other models, the authors demonstrate that SafeText achieves a high NRR on various unsafe prompt datasets while preserving image quality for safe prompts, outperforming six existing alignment methods.

**Strengths:**

1. The problem of harmful content generation in text-to-image models is both timely and important, especially as these models become more accessible and integrated into consumer applications. The authors rightly identify the trade-off between safety and utility as a key challenge in model alignment.

2. Most prior alignment approaches focus on modifying the diffusion module, which often degrades image quality for safe prompts. SafeText instead modifies the text encoder, which is a novel angle and potentially less disruptive to the image generation process.

3. SafeText consistently achieves the highest NRR across a wide range of unsafe prompt datasets while simultaneously recording the lowest utility degradation (as measured by LPIPS, FIDg, etc.) compared to a strong set of six recent baselines.

**Weaknesses:**

1. The paper argues that modifying the diffusion module leads to degradation in image quality for safe prompts. However, this argument is not quantitatively substantiated with sufficient empirical evidence. It is suggested to provide direct comparisons of image quality degradation across methods, showing how much more the diffusion module-based methods affect safe prompts compared to SafeText. Without such comparisons, the core motivation of the method is weakened.

2. The paper equates "safety" almost exclusively with the absence of nudity and sexually explicit content. The NRR metric is based on NudeNet, and the unsafe datasets are all focused on this domain. While the authors include a single short paragraph (lines 463-474) and a limited experiment on violence, this feels like an afterthought. Major categories of harmful content, such as extreme gore, self-harm imagery, promotion of hate ideologies, and generation of harmful misinformation, are not addressed in the evaluation. The title's broad claim of creating "Safe Text-to-Image Models" is not supported by the narrow scope of the experiments.

3. The core technical contribution is the formulation of the optimization problem in Equation (3): min (L_u - \lambda L_e). This approach, which aims to minimize the distance for "positive" pairs (safe prompts) while maximizing it for "negative" pairs (unsafe prompts), is conceptually similar to contrastive learning or triplet loss formulations that are well-established in machine learning. While the application to text encoder alignment is novel and effective, the underlying technical mechanism is not particularly groundbreaking, which may limit its conceptual contribution.

4. The choice of Euclidean distance for utility (d\_u) and negative cosine similarity for effectiveness (d\_e) appears arbitrary and is justified only through empirical ablation studies. The paper lacks theoretical analysis explaining why these particular distance metrics are optimal for their respective objectives. More concerningly, there's no discussion of how these distance choices relate to the actual geometry of the embedding space or the diffusion process. The empirical justification in Figures 16c-16d, while interesting, doesn't substitute for a proper theoretical framework.

**Questions:**

Refer to weaknesses

---

### Note · Authors · 2025-11-12

I have read and agree with the venue's withdrawal policy on behalf of myself and my co-authors.